physiology/neuroscience/health and disease and epidemiology

Monte Carlo, polycystic ovary syndrome, androgen, puberty, action potential

**Author for correspondence:**
Suzanne M. Moenter
e-mail: smoenter@umich.edu

# Firing patterns of gonadotropin-releasing hormone neurons are sculpted by their biologic state

Jonathon Penix[1], R. Anthony DeFazio[1], Eden A. Dulka[1], Santiago Schnell[1,2] and Suzanne M. Moenter[1,3,4]

[1]Departments of Molecular and Integrative Physiology, [2]Computational Medicine and Bioinformatics, [3]Internal Medicine, and [4]Obstetrics and Gynecology, University of Michigan, Ann Arbor, MI 48109, USA

 JP, 0000-0003-0643-7855; RAD, 0000-0001-7302-7528;
EAD, 0000-0002-7724-8923; SS, 0000-0002-9477-3914;
SMM, 0000-0001-9812-0497

Gonadotropin-releasing hormone (GnRH) neurons form the final pathway for the central neuronal control of fertility. GnRH is released in pulses that vary in frequency in females, helping drive hormonal changes of the reproductive cycle. In the common fertility disorder polycystic ovary syndrome (PCOS), persistent high-frequency hormone release is associated with disrupted cycles. We investigated long- and short-term action potential patterns of GnRH neurons in brain slices before and after puberty in female control and prenatally androgenized (PNA) mice, which mimic aspects of PCOS. A Monte Carlo (MC) approach was used to randomize action potential interval order. Dataset distributions were analysed to assess (i) if organization persists in GnRH neuron activity *in vitro*, and (ii) to determine if any organization changes with development and/or PNA treatment. GnRH neurons in adult control, but not PNA, mice produce long-term patterns different from MC distributions. Short-term patterns differ from MC distributions before puberty but become absorbed into the distributions with maturation, and the distributions narrow. These maturational changes are blunted by PNA treatment. Firing patterns of GnRH neurons in brain slices thus maintain organization dictated at least in part by the biologic status of the source and are disrupted in models of disease.

# 1. Introduction

Reproduction is controlled by interactions among the brain, anterior pituitary and gonads. Gonadotropin-releasing hormone (GnRH) neurons in the ventral diencephalon secrete GnRH near the hypothalamo-pituitary portal vasculature [1–3]. GnRH induces the anterior pituitary to synthesize and secrete luteinizing hormone (LH) and follicle-stimulating hormone (FSH) [4,5]. These hormones activate the gonadal functions of gametogenesis and steroidogenesis. Steroids provide feedback regulation to modulate the GnRH release pattern.

Inappropriate patterns of GnRH release can cause infertility [6,7]. One example is polycystic ovary syndrome (PCOS), the leading cause of infertility in women. Diagnosis requires two of the following three symptoms: absent/infrequent ovulation, elevated androgens and/or polycystic ovarian morphology [8]. Despite its high incidence (approx. 20% of women [9]), the root causes of PCOS are unknown. The present work is guided by findings that women with hyperandrogenemic PCOS, about 50% of affected women [7,10], have persistently high-frequency LH release, indicative of high-frequency GnRH release [11–13]. Action potential firing in neuroendocrine neurons like GnRH neurons is correlated with hormone release [14,15], thus a greater understanding of the activity of GnRH neurons in healthy versus PCOS states may identify mechanisms underlying increased GnRH/LH release frequency in PCOS.

Investigations of GnRH neuron physiology are not possible in humans. Prenatal exposure to androgens is a commonly used animal model to study this disorder. Prenatally androgenized (PNA) mice, rats, sheep and primates exhibit phenotypes that are similar to symptoms of women with PCOS, including disrupted reproductive cycles, increased androgens, and high LH pulse frequency [16–19]. Further, prenatal exposure to anti-Müllerian hormone (AMH) in mice elevates maternal neuroendocrine drive, resulting in increased prenatal androgen exposure and similar outcomes; AMH is elevated during gestation in women with PCOS [20]. While ovarian morphology aspects of PCOS in the rodent species have limitations [21], probably attributable to the polyovulatory nature of these species, the neuroendocrine phenotypes studied here are strikingly similar among species [22] and to the elevated LH pulse frequency observed in women [11–13]. In brain slices from adult PNA female mice [16], overall GnRH neuron firing rate is increased compared with controls, consistent with elevated LH pulse frequency observed *in vivo* [23]. Before puberty, however, firing rate is reduced in cells from PNA mice [23–25]. These observations suggest the postulate that PNA programmes a different developmental trajectory for GnRH neurons that results in different action potential firing output.

This research had two main goals. First, to address the overarching question of the validity of using GnRH neuron firing activity data to understand biological mechanisms, specifically if short- and/or long-term GnRH neuron firing activity is organized in a non-random manner. While GnRH release *in vivo* is clearly organized into discrete pulses [1–3], these occur in the context of the whole animal's physiology. Making brain slices removes both peripheral and central inputs to GnRH neurons that may contribute to this organization. Work has examined if physiologic state of the originating animal alters the fairly simple measure of mean firing rate of these cells in brain slices [20,26–30]. More formal investigations of pattern organization are limited [31–33], however, and whether or not these patterns differ from a distribution of possible datasets with the same inter-event intervals has not been examined. The second goal was to determine if elements of pattern organization differ with reproductive state. To achieve these goals, we used Monte Carlo (MC) randomization [34] to perform additional analyses on a subset of the data from Dulka and Moenter [25]. MC is a quantitative method that generates multiple randomizations of an aspect of the original dataset (here, intervals) to create a distribution of possibilities to which the original dataset can be statistically compared. Randomization tests like Monte Carlo analyses provide power by reducing the constraints of small samples, distributions and unequal variance. These methods do not make assumptions about data distribution and can be used to address questions such as how likely it is to achieve a specific outcome.

# 2. Material and methods

## 2.1. Data used

Data were from [25]; collection of those data was approved by the Institutional Animal Care and Use Committee of the University of Michigan (protocol 6816). We focused on female control and PNA mice at three weeks of age (3wk) and in adulthood (adult, 17–38 wks). These groups were chosen as they exhibited the greatest difference in mean firing rate, interspike intervals and burst patterning and thus made interesting points to examine if GnRH neuron firing activity is organized, and if any organization changes with development or disease model.

Action currents, the currents associated with action potentials, were recorded from green-fluorescent-protein-identified neurons in acutely prepared brain slices [25]. Action currents (events) were detected and confirmed, and inter-event intervals obtained. These intervals, and the MC randomizations thereof (below), were analysed in two ways. The Cluster algorithm examines what we define as long-term patterns. Cluster was originally designed to find patterns in the release of hormones such as LH [35], which is typically sampled at 5 to 10 min intervals from the blood of experimental subjects. Electrophysiological data are collected at sub-millisecond intervals and are thus oversampled for this analysis aimed to detect long-term patterns that may be associated with hormone release. To format our data more suitably for use with the Cluster algorithm, data were divided into 120 s bins. Multiple sequential bins are compared with one another to detect peaks and nadirs in firing rate. The vary burst window (VBW) algorithm examines what we define as short-term patterns. This algorithm works on raw intervals, determining if an event can be grouped with the previous event based on a user-determined inter-event interval [25,36–38]. VBW automates iterative changes in the user-defined interval, and groups of events are referred to as bursts.

## 2.2. Monte Carlo approach

To address whether or not short- and/or long-term GnRH neuron firing activity is organized in a non-random manner, we used a Monte Carlo approach in which we generate random permutations of the original data to create surrogate datasets (referred to as MC datasets below). Because we were interested in the organization of action potentials, we randomized the order of event intervals. This approach has the advantage that we do not need to have *a priori* knowledge about the underlying distribution of GnRH neuron firing activity with these interval characteristics. Using the surrogate data, we are then able to test the null hypothesis that the firing intervals and the resulting groupings of events observed (individual spikes, bursts and clusters) are generated by chance.

## 2.3. Permutation generation

To generate the permuted data for the MC approach, the ordering of the intervals between events of each original recording was randomized using a version of Durstenfeld's shuffle algorithm [39] implemented in IgorPro. Random values were obtained through the language's standard library functions using the ran2 algorithm as the underlying pseudo-random number generator [40]. This process was repeated 1000 times for each cell [41,42]. Original data and the 1000 randomized MC datasets were subjected to the two analyses described above. Limitations to MC approaches include: (i) readily available pseudo-random number generators cannot generate all possible permutations for all of the cells studied, (ii) 1000 is lower than the maximum permutations possible in the data being analysed, and (iii) the number of permutations is different among the biological groups examined because action potential frequency is altered, leading to different numbers of intervals in similar length recordings.

## 2.4. Long-term patterns

*In vivo*, GnRH pulse frequency is modulated by steroids and varies from once every several minutes to once every few hours during the female reproductive cycle [43,44]. Frequency can also change in response to experimental manipulation [45–47], to disease states such as PCOS [11–13] or to natural changes in fertility such as in seasonal breeders [48]. Peaks and nadirs within the firing rate data are of interest as they are hypothesized to be associated with neurosecretion, based on their interval being similar to that of LH pulses *in vivo* [31,32]. Peaks were identified using a version of the Cluster algorithm [35] implemented in IgorPro [37]. Based on previous studies of LH and GnRH pulses and GnRH neuron activity [31,32,44,49], Clusters of $2 \times 2$ bins (for peaks and nadirs, respectively) with a t-score of 2 were used to identify increases and decreases, and the local standard deviation was used to estimate error. Proximal variations in these parameters (cluster sizes of 1–5, t-scores of 1–3, local versus global errors) were tested but did not alter the outcomes. Cluster output parameters analysed include number of peaks in firing rate, peak frequency, amplitude and duration. These outputs and the methodology are well described and illustrated in the following reviews [50,51].

## 2.5. Short-term patterns

Whereas Cluster analysis gives insight into the long-term organization of the cell's activity, burst properties characterize comparatively short-term patterns within the data. Bursts are groups of action potentials separated by short (millisecond to second) intervals and are often characteristic of neurons

[33,52–58]. Bursts are of interest as they are correlated with neurosecretion [14,15]. Bursts were detected using burst windows from 0 to 2000 ms in 10 ms increments. Based on the distribution of number of bursts detected (see below), most analyses were confined to burst windows at 150 ms intervals from 60 ms to 810 ms as these span the burst windows at which the highest number of bursts were detected, as well as intervals used for previous studies of GnRH pattern analysis (210 ms and 360 ms, [25,53]). Burst parameters analysed include burst frequency, spikes/burst, burst duration, inter-event interval (bursts and single spikes included as events), intraburst interval and single-spike frequency.

## 2.6. Analyses

Cluster and VBW algorithms were run as described on each of the 1000 MC datasets, yielding frequency distributions for the output parameters (Cluster: frequency amplitude and duration of elevated firing; VBW: burst frequency, inter-event and intra-event intervals, burst duration, spikes/burst, single-spike frequency). Given the large number of datasets and their randomized nature, we assume these distributions are representative of the underlying probability distribution of data with these interval characteristics. Because these analyses account for the original data as well as the MC datasets, we refer to these as total datasets (1000 MC runs plus original data). Initially, the proportions of total datasets with a parameter greater than or equal to versus less than or equal to the corresponding parameter of the original data were calculated to test if the original falls towards the upper or lower tails of the distribution, respectively. The inclusion of values that were equal to the result from the original data in these calculations revealed a high percentage of overlap for some parameters, particularly in the number of peaks detected with Cluster (attributable to their whole number nature and limited number of possible values), as well as burst parameters at burst windows that were shorter than the majority of the observed intervals within the original data. Datasets were thus divided into three additional classifications: those with the parameter being less than that of the original, those with parameters equal to that of the original and those with parameters greater than that of the original. Because these analyses include original data plus the MC datasets, the label for the ordinate was changed from %MC datasets to %total datasets.

Additional analyses were then performed to quantify the relationship between the original data and the generated frequency distributions, and to determine if there were differences among treatment groups.

## 2.7. Permutation tests

Permutation tests estimate the likelihood that the order of inter-event intervals in the original data is arbitrary by comparing it to a distribution estimated by the MC datasets [59–62]. This approach is non-parametric and does not require assumptions about whether or not the distributions are normal or had equal variance. Permutation tests were thus used to determine if two sets of data have different distributions for any of the parameters quantified by Cluster or VBW analysis. Random values used in the permutation tests were generated using Python's standard library functions for generating random numbers, using Mersenne twister [63] as the underlying pseudo-random number generator. These tests were independently run using medians and means. Results were similar and we chose to use medians because not all data were normally distributed, indicating non-parametric methods are more appropriate.

In brief, to generate permutations, the difference of medians for two sets $A$ and $B$ was calculated and the observations from each set combined. Two new sets $A'$ and $B'$ (the size of $A$ and $B$, respectively) were then constructed by randomly assigning each observation to one of the sets. The difference of medians for $A'$ and $B'$ were recorded, and this process repeated 1000 times. The number of times the difference of medians for the generated sets $A'$ and $B'$ were greater than or equal to the difference of medians for $A$ and $B$ was divided by 1000 to obtain a true two-tailed $p$-value.

## 2.8. Cell versus itself permutation test

To determine if the original data differ from the distribution of MC datasets generated from it, we used the above permutation test method, taking the original data and 1000 MC runs from each cell as the two sets to be compared. In this design, we test the null hypothesis that intervals in the original data are random.

## 2.9. Pairwise group permutation test

To generate $p$-values for comparing effects of age and treatment, the above permutation method was performed between the development and treatment groups (i.e. all 3wk versus all adult; all control versus all PNA). Logical individual pairwise comparisons within this two-by-two design (e.g. adult control versus adult PNA) were then compared (i.e. $<$, $>$, $=$, $\leq$, $\geq$ corresponding original data) when justified. In this design, we test the null hypotheses that age and/or treatment does not affect the proportion of MC datasets that are $<$, $>$, $=$, $\leq$, $\geq$ the corresponding original data for any of the parameters quantified by Cluster or VBW analysis.

## 2.10. Binomial exact tests

To identify trends within a group for the number of cells with differences to the MC distribution as determined by the cell versus itself permutation tests, binomial exact tests were used to compare the proportions of cells in each group with $p$-values less than 0.05 and with $p$-values $\geq 0.05$ to the proportion of values in groups of this size that would be expected to fall into these categories if distribution were random. Expected proportion for $p < 0.05$ was defined as $1/n$ for each group (range 7.7–14%), with the assumption being that typically one value in groups of these sizes would fall towards a tail of the distribution. This expectation was validated by performing the same cell versus itself analysis (100×) on a set of randomly generated datasets of similar value ranges; $p$-values were distributed fairly evenly across the possible range. Outcomes did not change when the expected proportion off cells with $p$-values less than 0.05 was set to 10% for all groups, thus the variation between 7.7 and 14% did not affect outcomes. Binomial exact tests are suitable for this comparison as they allow for the testing of deviations from an expected distribution of observations into two categories and are not subject to the same limitations on sample sizes as other related tests (Chi-squared test).

# 3. Results

## 3.1. Do long-term firing patterns of gonadotropin-releasing hormone neurons differ from the Monte Carlo distribution?

To examine if the firing patterns generated by GnRH neurons differ from random, the cell versus itself analysis was used to compare the original data from each individual cell to its own MC datasets. Figure 1$a$–$d$ shows the distribution of number of peaks detected by Cluster analysis of the 1000 MC datasets from four representative recordings from each group (black bars) and the position of the original data (magenta line). The $p$-values from the cell versus itself analysis for each cell for the Cluster parameters examined are shown in figure 1$e$. In cells from control mice, the original data had fewer peaks in firing rate than in the MC datasets and are thus located towards the left end of the distribution in most examples in figure 1$a$ and $c$. By contrast, in PNA mice original data were typically within the distributions (figure 1$b$,$d$). Cell versus itself analyses also indicated differences between the original data and distributions for amplitude and duration of these peaks in firing rate (figure 1$e$). In three-week-old and adult vehicle (VEH) control mice, a higher than expected proportion of cells had original data that were different from its MC distribution for both frequency and duration of peaks in firing rate (binomial exact test, table 1). Amplitude was also different from the MC distribution in all groups except adult vehicle.

The relationship of the MC datasets to the original data for Cluster-detected peak frequency, amplitude and duration is shown in figure 2. Each dot shows the percentage of total datasets from a cell that is less than, equal to or greater than the original data's value for that particular parameter. Unlike the cell versus itself permutation test, this analysis reveals not only if the original data are different from the distribution of MC values for a particular parameter, but also the direction of change. This analysis supports the above conclusion that the long-term patterns arising from the original data exhibit fewer, longer duration peaks than the MC distributions. For amplitude, however, differences revealed by binomial exact tests may be attributable to the spread of data, rather than a specific directional shift as was observed for peak frequency and duration. These observations suggest long-term GnRH neuron activity is organized to generate fewer,

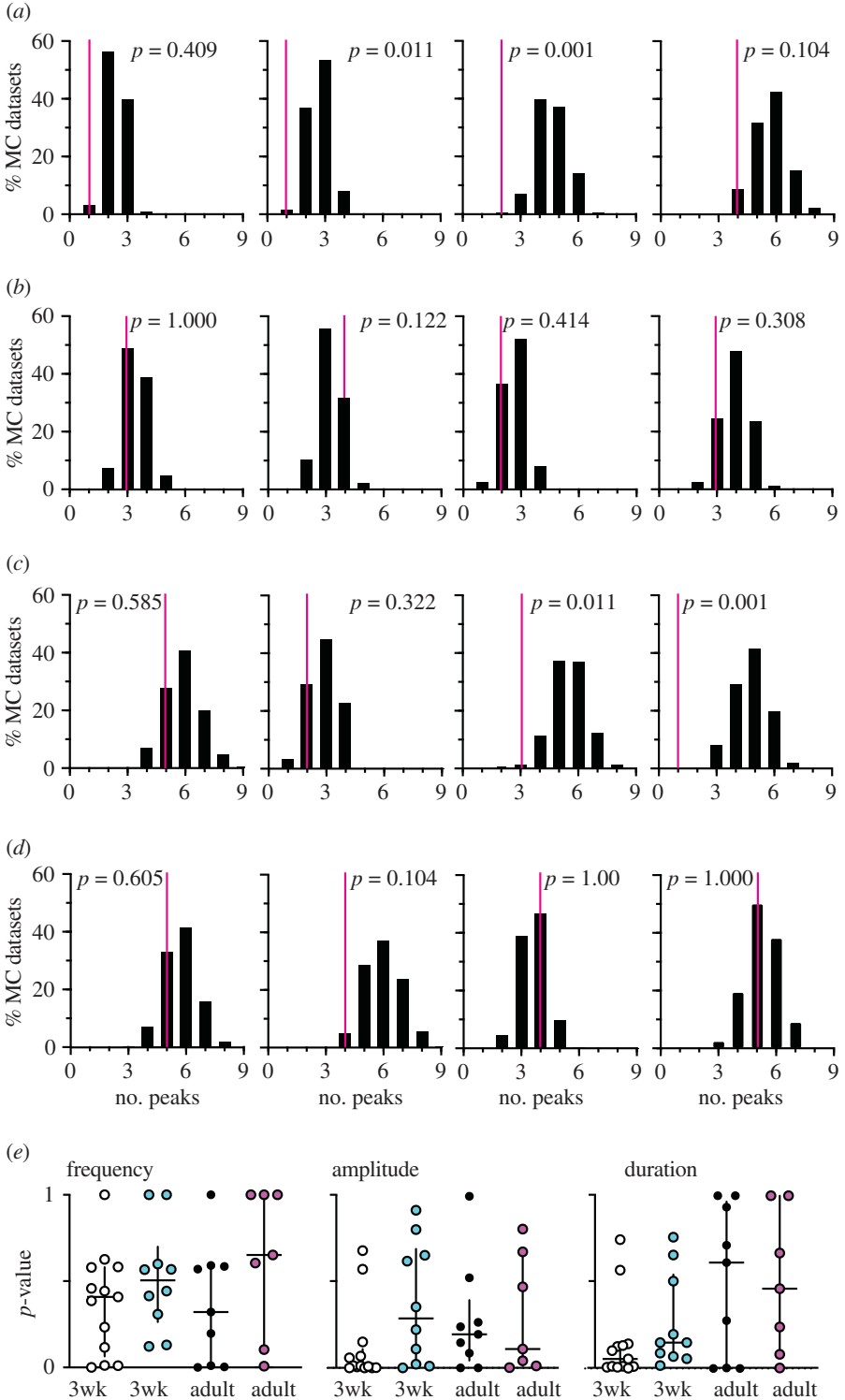

**Figure 1.** Original data (magenta line) versus MC distribution for four representative recordings from each group. (*a*–*d*) Representative examples of the distribution of number of peaks detected by Cluster in the 1000 MC datasets. *p*-values indicated are from cell versus itself analysis comparing original data to its MC dataset distribution. (*a*) three-week-old vehicle; (*b*) three-week-old PNA; (*c*) adult vehicle; (*d*) adult PNA. (*e*) *p*-values for each cell from cell versus itself analysis comparing original data to its MC distribution for frequency of peaks (left), peak amplitude (centre) and peak duration (right).

longer peaks than random activity, and that PNA treatment disrupts this organization both before and after puberty; we thus reject the null hypothesis that intervals in the original data are random for long-term patterns.

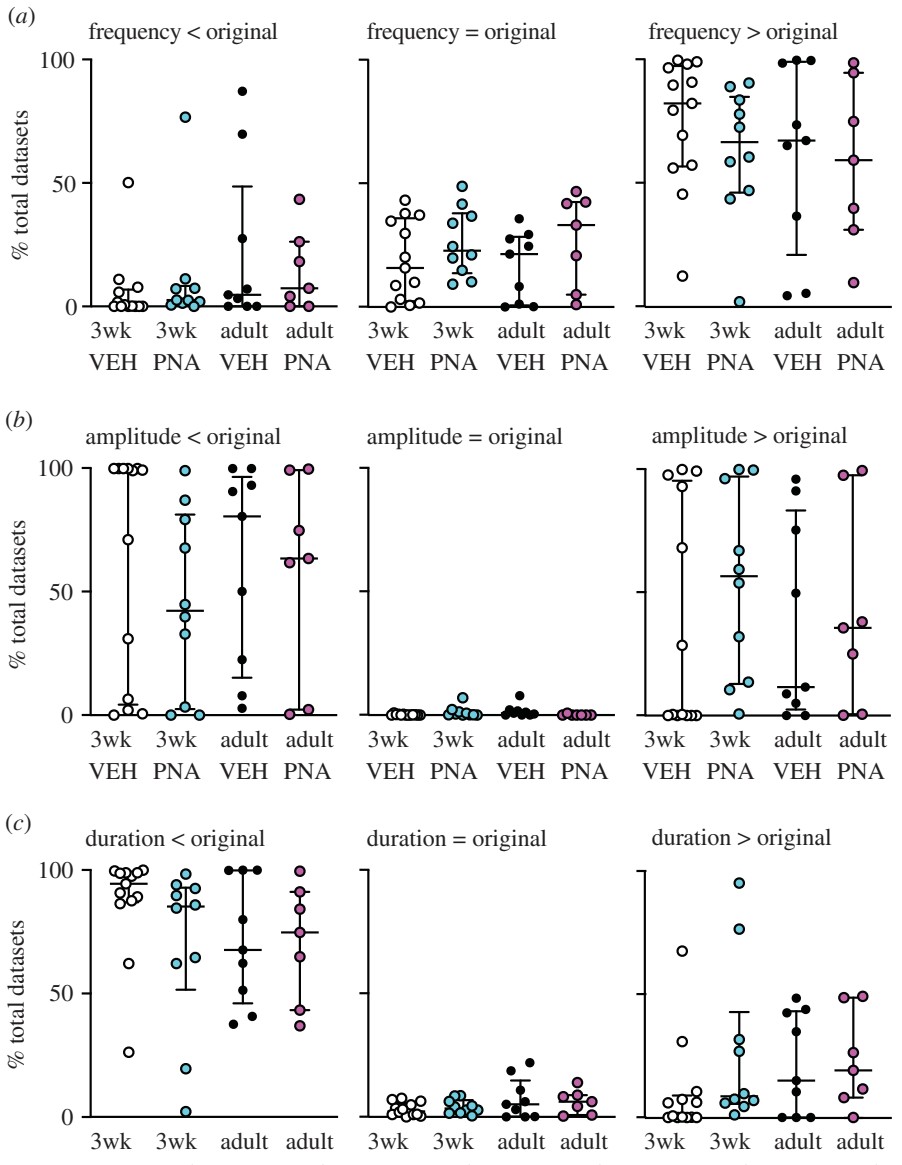

**Figure 2.** GnRH neurons lower the frequency and increase duration of long-term peaks in action potential firing. Cluster analysis of MC datasets for peak frequency (*a*), amplitude (*b*) and duration (*c*) compared with original data. Graphs show individual values, median and interquartile range. Each dot shows the percentage of total datasets from an individual cell that is less than (left), equal to (centre) or greater than (right) the original data's value for that parameter.

**Table 1.** *p*-values from binomial exact test comparing the proportion of cells that were different from their MC distributions by cell versus itself permutation tests to the expected proportion for Cluster analysis outputs. Italics, $p \leq 0.05$; bold, $0.05 < p < 0.1$.

| peak parameter | 3wk vehicle ($n = 13$) | 3wk PNA ($n = 9$) | adult vehicle ($n = 9$) | adult PNA ($n = 7$) |
|---|---|---|---|---|
| frequency | **0.073** | 0.612 | **0.069** | 1 |
| amplitude | *$1.11 \times 10^{-6}$* | **0.070** | 0.264 | **0.065** |
| duration | *0.001* | 1 | **0.069** | 1 |

## 3.2. Do age and/or prenatal androgen treatment affect long-term firing patterns of gonadotropin-releasing hormone neurons?

In the original experiment [25], Cluster analysis to assess differences among groups in long-term firing pattern was not performed. This analysis revealed no difference in number of peaks in firing rate,

**Table 2.** p-values generated by pairwise group permutation tests of median values for main effects of age and treatment on Cluster-detected peaks in firing rate. Bold, $0.05 < p < 0.1$.

| Cluster parameter, relation of randomized (R) to original (O) data | p-values age | p-values treatment |
|---|---|---|
| frequency | | |
| R = 0 | 0.671 | 0.386 |
| R > 0 | 0.381 | 0.217 |
| R ≥ 0 | 0.205 | 0.359 |
| R < 0 | 0.217 | 0.347 |
| R ≤ 0 | 0.396 | 0.213 |
| amplitude | | |
| R = 0 | 0.550 | 1.000 |
| R > 0 | 0.934 | 0.393 |
| R ≥ 0 | 0.950 | 0.391 |
| R < 0 | 0.937 | 0.400 |
| R ≤ 0 | 0.949 | 0.381 |
| duration | | |
| R = 0 | **0.098** | 0.574 |
| R > 0 | **0.079** | 0.188 |
| R ≥ 0 | **0.081** | 0.364 |
| R < 0 | **0.067** | 0.391 |
| R ≤ 0 | **0.082** | 0.181 |

although the main effect of age approached the value traditionally set for significance (mean ± s.e.m.: 3wk vehicle 2.9 ± 0.5, PNA 3.8 ± 0.4; adult vehicle 4.1 ± 0.8, PNA 4.4 ± 0.3; main effect of age $F_{1,36} = 2.901$, $p = 0.0972$). Interestingly, the mode did increase with age (mean ± s.e.m.: 3wk vehicle 4.0 ± 0.4, PNA 4.3 ± 0.4; adult vehicle 5.1 ± 0.3, PNA 5.4 ± 0.3; main effect of age $F_{1,36} = 8.637$, $p = 0.0057$). Permutation tests also revealed that neither age nor treatment affected the rank of the original data versus the total data distribution of the Cluster parameters (table 2). A tendency for peak duration to be increased at three weeks of age was observed (combined 3wk versus combined adult) as p-values ranged from 0.06 to 0.1. Because there was no main effect of age or treatment, pairwise comparisons were not evaluated for Cluster parameters, and we accept the null hypothesis that age and/or treatment does not affect the proportion of MC datasets that are <, >, =, ≤, ≥ the corresponding original data for long-term patterns.

## 3.3. Burst window analysis of short-term firing patterns

MC datasets were next examined using the VBW algorithm. How VBW identifies bursts is shown in figure 3a and the relationship between interval duration and number of occurrences in the original datasets is in figure 3b. As burst window increases in duration, the number of bursts detected increases to a peak, then declines as bursts are merged with one another (figure 3c). The median per cent of total datasets with different relationships (<, =, >, ≤, ≥) to the original data for burst frequency is shown for all burst windows examined up to 1 s in figure 4. Original data from adult vehicle controls did not exhibit any differences from the corresponding total MC dataset distributions. This is in contrast with the other three groups, in which burst frequencies of most MC datasets were greater than in the original. This suggested the postulates that (i) burst frequency changes with typical development, and (ii) development of adult burst patterns is disrupted by PNA treatment, as adult PNA mice more closely resemble three-week-old mice from either group than adult vehicle controls.

## 3.4. Do short-term firing patterns of gonadotropin-releasing hormone neurons differ from the Monte Carlo distribution?

To examine these postulates in a more rigorous manner, selected burst windows (every 150 ms from 60 to 810 ms, figure 3c) were examined as above. VBW parameters of a cell's original data were first

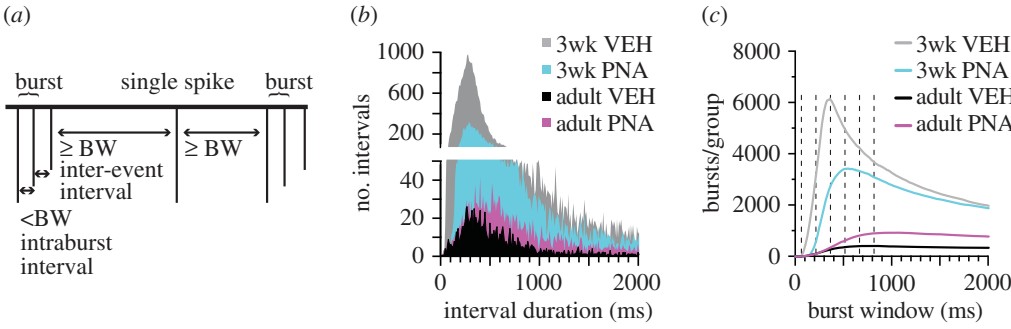

**Figure 3.** (*a*) VBW algorithm uses an iterative window to examine time-series data. When a subsequent event occurs within less than the duration of the burst window (BW), it is combined with the previous event(s) into a burst. (*b*) Number of intervals as a function of interval duration. (*c*) Total number of bursts/group as a function of burst window for the original datasets. Dashed lines at 60, 210, 360, 510, 660 and 810 ms indicate burst windows chosen for analyses.

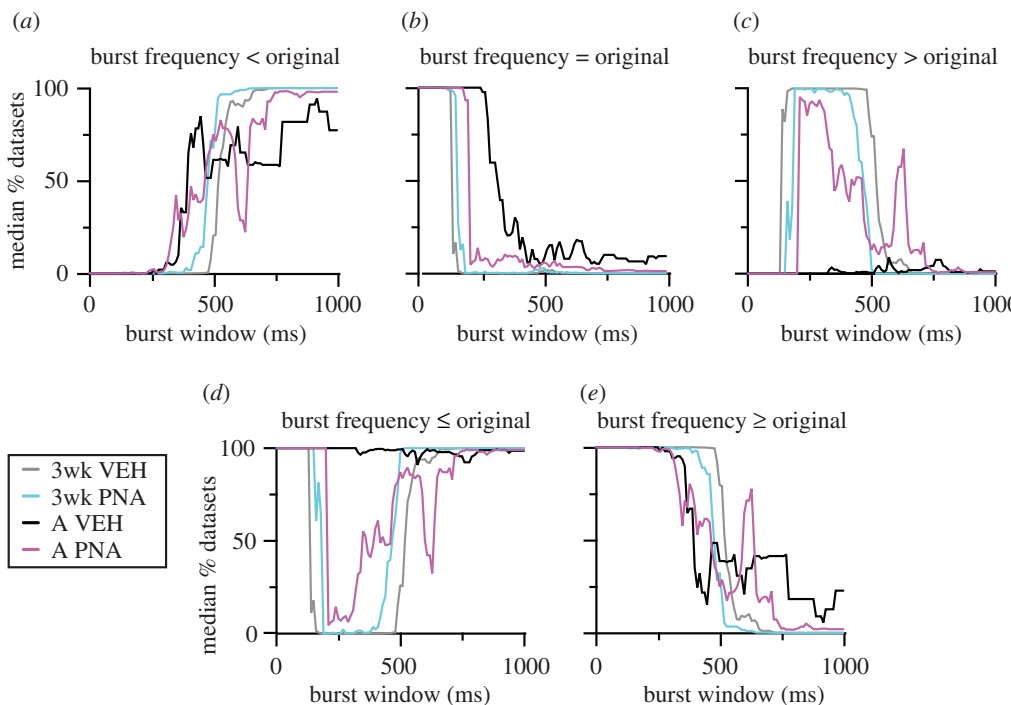

**Figure 4.** Median per cent of datasets from each group that are less than original data (*a*), equal to the original data (*b*), greater than the original data (*c*), less than or equal to the original data (*d*) and greater than or equal to the original data (*e*) as a function of burst window. 3wk, three weeks of age; A, adult.

compared with those of its MC datasets. Binomial exact tests revealed a higher than expected proportion of cells for which the cell versus itself permutation tests demonstrated that the original data were different from its MC distribution for several burst parameters (table 3). In the physiologic range of burst windows (210, 360 ms [25,53]), GnRH neurons from most groups generated fewer, longer bursts and had fewer single spikes than the MC datasets (figure 5). Consistent with the first postulate above, the exception was cells from adult vehicle (VEH) mice, for which the original data were not different from the MC datasets for any parameter at 210 or 360 ms burst windows. This age-dependent shift in control mice was associated with a narrowing of the range (maximum minus minimum) of the data distributions (mean ± s.e.m.: 3wk control $65.3 \pm 11.9$, PNA $44.3 \pm 8.7$; adult vehicle control $9.9 \pm 3.8$, PNA $16.1 \pm 3.2$; main effect of age $F_{1,36} = 18.66$, $p < 0.0001$). This suggests the intervals extant in the original datasets from adult controls constrains the range of possibilities for bursts. Of note, only four cells in the adult vehicle control group exhibited bursts as defined by the 210 ms window, this lowered the number of observations for burst characteristics to unacceptable levels for analysis for parameters other than burst frequency, single-spike frequency and inter-event interval. Together these

**Table 3.** p-values from binomial exact test comparing the proportion of cells that were different from their MC distributions by cell versus itself permutation tests to the expected proportion for VBW analysis outputs. Italics, $p \leq 0.05$; bold, $0.05 < p < 0.1$.

| 210 ms burst window parameter | 3wk VEH (n = 13) | 3wk PNA (n = 9) | adult VEH (n = 9) | adult PNA (n = 7) |
|---|---|---|---|---|
| burst frequency | $1.80 \times 10^{-5}$ | $1.22 \times 10^{-5}$ | 0.610 | 0.010 |
| inter-event interval | $1.80 \times 10^{-5}$ | 0.002 | 1 | **0.065** |
| intraburst interval | 1 | 0.614 | | 0.603 |
| burst duration | $5.98 \times 10^{-7}$ | $1.22 \times 10^{-5}$ | | 0.010 |
| spikes/burst | $5.98 \times 10^{-7}$ | $1.22 \times 10^{-5}$ | | 0.010 |
| single-spike frequency | $1.80 \times 10^{-5}$ | 0.001 | 1 | 0.264 |
| **360 ms burst window parameter** | **3wk VEH (n = 13)** | **3wk PNA (n = 9)** | **adult VEH (n = 9 or 8)** | **adult PNA (n = 7)** |
| burst frequency | $3.76 \times 10^{-11}$ | $5.98 \times 10^{-7}$ | 1 | 0.264 |
| inter-event interval | 0.001 | 0.002 | 1 | 0.010 |
| intraburst interval | 1 | 1 | 1 | 0.603 |
| burst duration | $3.76 \times 10^{-11}$ | $5.98 \times 10^{-7}$ | 1 | 0.264 |
| spikes/burst | $3.7 \times 10^{-11}$ | $5.98 \times 10^{-7}$ | 1 | 0.264 |
| single-spike frequency | $3.76 \times 10^{-11}$ | $5.98 \times 10^{-7}$ | 1 | 0.264 |
| **510 ms burst window parameter** | **3wk VEH (n = 13)** | **3wk PNA (n = 9)** | **adult VEH (n = 9)** | **adult PNA (n = 7)** |
| burst frequency | 0.002 | 0.001 | 0.012 | **0.065** |
| inter-event interval | 0.001 | 0.002 | 1 | **0.065** |
| intraburst interval | 1 | 0.614 | 1 | 0.603 |
| burst duration | 0.002 | 0.002 | 0.264 | **0.065** |
| spikes/burst | 0.002 | 0.002 | 0.264 | **0.065** |
| single-spike frequency | 0.001 | 0.002 | 0.012 | **0.065** |

observations suggest the developmental trajectory of short-term firing is different in VEH and PNA mice. This results in a split decision on the null hypothesis that intervals in the original data are random for short-term patterns, with acceptance for adult controls but rejection for other groups.

## 3.5. Do age and/or treatment affect short-term firing patterns of gonadotropin-releasing hormone neurons?

Comparing the effects of age and treatment on the MC distributions of these VBW parameters using pairwise group permutation tests revealed an effect of age on burst frequency, duration, spikes/burst and single-spike frequency but no effect on inter-event or intraburst interval (table 4 and figures 6–8). These differences were observed in the physiologic range of 210–360 ms burst windows. The 60 ms burst window failed to detect any bursts, not surprising as this is shorter than most intervals between action potentials in GnRH neurons. Differences with age were largely lost at the 510 ms burst window, reappearing at 660 and 810 ms. These three burst windows are past the peak of interval durations (figure 3a) and past the peak of bursts/group as a function of burst window (figure 3c). Pairwise analysis was thus confined to 210, 360 and 510 ms burst windows. These analyses revealed differences between 3wk and adult VEH mice for burst frequency, duration, spikes per burst and single-spike frequency (supporting the first postulate), and between adult VEH and adult PNA for burst frequency and duration (supporting the second postulate, table 5). We thus reject the null hypothesis that age and/or treatment does not affect the proportion of MC datasets that are <, >, =, ≤, ≥ the corresponding original data for long-term patterns.

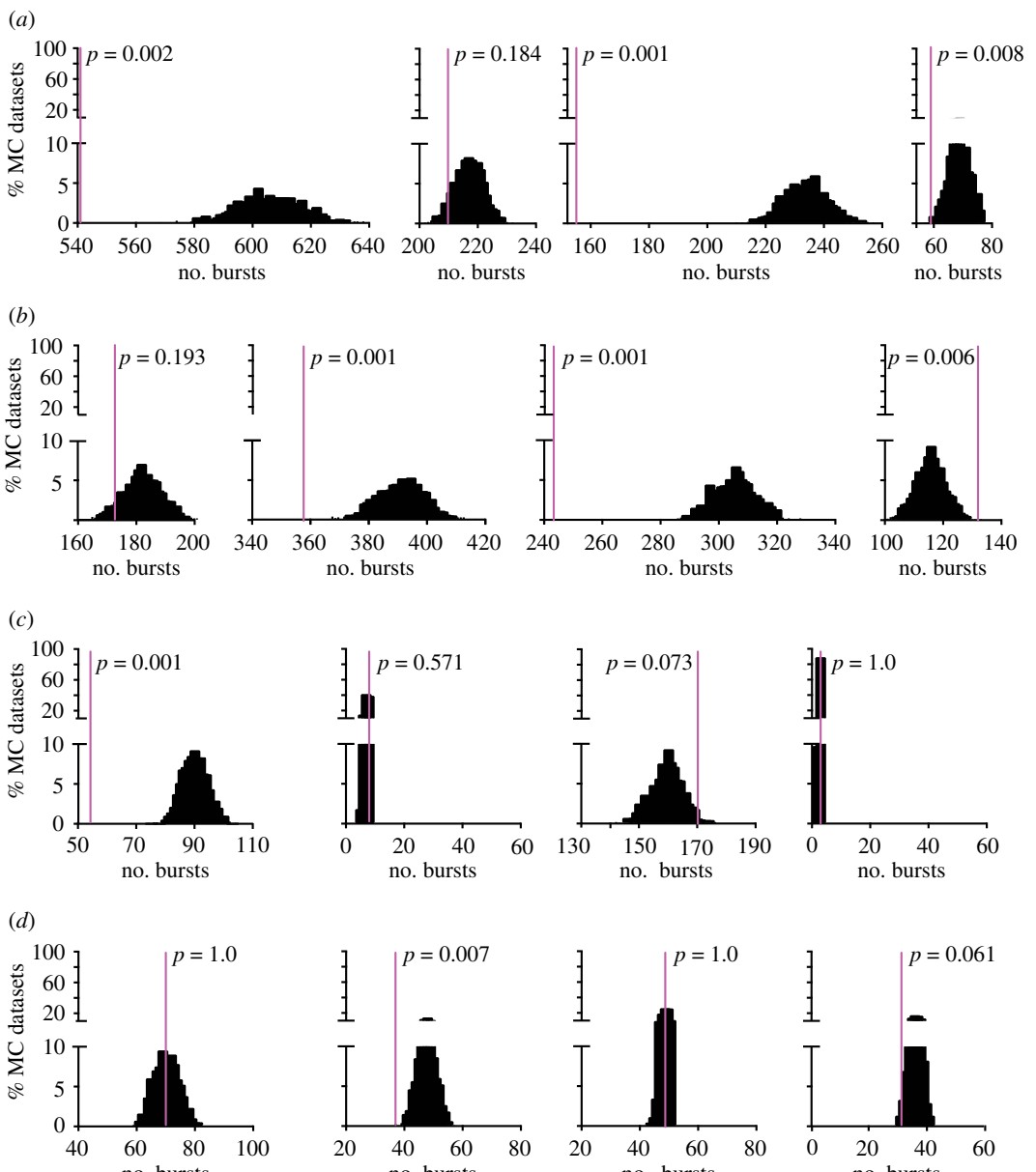

**Figure 5.** Representative examples from each group of the distribution of the number of bursts detected by the VBW algorithm with a window of 360 ms in the 1000 MC datasets versus original data (magenta line). (*a*) three-week-old vehicle; (*b*) three-week-old PNA; (*c*) adult vehicle; (*d*) adult PNA. *p*-values are from cell versus itself analysis comparing original data to its MC dataset distribution. X-axis scale is consistent; range of values differs.

## 4. Discussion

Episodic, frequency-modulated GnRH release is critical for fertility. *In vivo*, this output is typically monitored as LH release frequency, as direct monitoring of GnRH requires access to the vasculature connecting the brain to the pituitary. In reduced preparations, such as brain slices used for electrophysiology, other parameters such as action potential firing rate become available for study. Here, we used MC analysis to examine both short- and long-term patterns in GnRH neuron firing rate. The present studies revealed that GnRH neurons have lower frequency, longer duration bouts of firing activity for both short- and long-term firing patterns than would be expected if activity was random. They further revealed that elements of these patterns change with age, and that this maturation is incomplete in PNA mice.

The Cluster algorithm was used to study long-term firing patterns. Cluster was originally designed to analyse the patterns in serum hormone levels and has been used to examine LH [35] and GnRH [44]

**Table 4.** *p*-values generated by pairwise group permutation tests of median values for main effects of age and treatment on bursts detected by vary burst window. O, original data; spb, spikes/burst; ssf, single-spike frequency; inter-event, inter-event interval; intraburst, intraburst interval. Italics, $p \leq 0.05$; bold, $0.05 < p < 0.1$.

| parameter, relation of randomized (R) to original (O) data ↓ | 60 ms | | 210 ms | | 360 ms | | 510 ms | | 660 ms | | 810 ms | |
|---|---|---|---|---|---|---|---|---|---|---|---|---|
| age (A) or treatment (T) → | A | T | A | T | A | T | A | T | A | T | A | T |
| frequency | | | | | | | | | | | | |
| R = 0 | 1 | 1 | *0.017* | 0.362 | *0.003* | 0.936 | *0.017* | 0.985 | *0.003* | 0.962 | *0.003* | 0.749 |
| R > 0 | 1 | 1 | *0.003* | 0.39 | *0.05* | 0.935 | 0.225 | 1 | **0.091** | 0.775 | **0.077** | 0.329 |
| R ≥ 0 | 1 | 1 | 0.533 | 0.517 | *0.036* | 0.542 | 0.718 | 0.27 | *0.035* | 0.211 | **0.096** | 0.231 |
| R < 0 | 1 | 1 | 0.503 | 0.542 | *0.033* | 0.503 | 0.735 | 0.292 | *0.025* | 0.245 | 0.112 | 0.226 |
| R ≤ 0 | 1 | 1 | *0.001* | 0.368 | **0.058** | 0.934 | 0.235 | 1 | **0.071** | 0.745 | *0.077* | 0.377 |
| duration | | | | | | | | | | | | |
| R = 0 | 0.318 | 0.716 | *0.048* | 0.678 | *0.004* | 1 | *0.046* | 0.718 | *0.005* | 1 | *0.003* | 0.497 |
| R > 0 | 1 | 1 | 0.126 | 1 | **0.077** | 0.384 | 0.553 | 0.249 | *0.028* | 0.403 | **0.059** | 0.233 |
| R ≥ 0 | 0.281 | 0.83 | *0.014* | 0.977 | *0.031* | 0.878 | 0.347 | 0.691 | **0.089** | 0.385 | **0.056** | 0.137 |
| R < 0 | 0.33 | 0.862 | *0.019* | 0.967 | **0.051** | 0.885 | 0.311 | 0.69 | **0.085** | 0.363 | **0.068** | 0.155 |
| R ≤ 0 | 1 | 1 | 0.113 | 1 | *0.045* | 0.392 | 0.576 | 0.272 | *0.024* | 0.394 | **0.076** | 0.222 |
| spb | | | | | | | | | | | | |
| R = 0 | 0.467 | 0.504 | *0.042* | 0.875 | *0.004* | 1 | 0.118 | 0.851 | *0.002* | 0.843 | *0.001* | 0.755 |
| R > 0 | 1 | 0.418 | 0.122 | 1 | **0.053** | 0.466 | 0.556 | 0.331 | *0.022* | 0.412 | **0.084** | 0.242 |
| R ≥ 0 | 0.472 | 1 | *0.024* | 0.659 | *0.045* | 0.891 | 0.375 | 0.842 | *0.049* | 0.519 | **0.075** | 0.255 |
| R < 0 | 0.469 | 1 | *0.023* | 0.685 | *0.05* | 0.886 | 0.366 | 0.838 | **0.066** | 0.537 | **0.075** | 0.236 |
| R ≤ 0 | 1 | 0.418 | 0.123 | 1 | *0.036* | 0.474 | 0.579 | 0.336 | *0.041* | 0.401 | **0.076** | 0.238 |

(*Continued.*)

**Table 4.** (*Continued.*)

| parameter, relation of randomized (R) to original (O) data ↓ | 60 ms | | 210 ms | | 360 ms | | 510 ms | | 660 ms | | 810 ms | |
| age (A) or treatment (T)→ | A | T | A | T | A | T | A | T | A | T | A | T |
|---|---|---|---|---|---|---|---|---|---|---|---|---|
| ssf | | | | | | | | | | | | |
| R = 0 | 1 | 1 | *0.02* | 0.422 | *0.004* | 0.879 | *0.044* | 0.861 | *0.001* | 0.947 | *0.005* | 0.736 |
| R > 0 | 1 | 1 | 0.295 | 0.212 | *0.03* | 0.631 | 0.631 | 0.291 | **0.067** | 0.266 | **0.099** | 0.244 |
| R ≥ 0 | 1 | 1 | *0.005* | 0.392 | *0.038* | 0.937 | 0.217 | 0.996 | **0.077** | 0.832 | **0.065** | 0.328 |
| R < 0 | 1 | 1 | *0.006* | 0.373 | *0.049* | 0.933 | 0.228 | 0.99 | **0.096** | 0.821 | **0.065** | 0.336 |
| R ≤ 0 | 1 | 1 | 0.286 | 0.218 | *0.028* | 0.617 | 0.668 | 0.289 | **0.052** | 0.284 | 0.101 | 0.233 |
| inter-event | | | | | | | | | | | | |
| R = 0 | **0.079** | 0.234 | *0.044* | 0.171 | **0.051** | 0.028 | *0.033* | **0.076** | **0.065** | **0.079** | **0.052** | 0.142 |
| R > 0 | 0.456 | 0.98 | 0.523 | 0.917 | 0.552 | 0.937 | 0.638 | 0.687 | 0.748 | 0.776 | 0.76 | 0.777 |
| R ≥ 0 | 0.47 | 1 | 0.48 | 0.868 | 0.347 | 0.927 | 0.277 | 0.739 | **0.067** | 0.659 | *0.046* | 0.604 |
| R < 0 | 0.494 | 1 | 0.463 | 0.855 | 0.382 | 0.928 | 0.258 | 0.72 | **0.072** | 0.643 | *0.049* | 0.651 |
| R ≤ 0 | 0.478 | 0.972 | 0.504 | 0.922 | 0.546 | 0.958 | 0.666 | 0.667 | 0.721 | 0.785 | 0.762 | 0.789 |
| intraburst | | | | | | | | | | | | |
| R = 0 | 0.299 | 1 | 0.449 | 0.186 | *0.027* | 0.918 | 0.305 | 0.415 | 0.184 | 0.61 | **0.051** | 0.924 |
| R > 0 | **0.052** | 0.42 | 0.624 | 0.364 | 0.141 | 0.746 | 0.115 | 0.909 | 0.342 | 0.433 | 0.29 | 0.524 |
| R ≥ 0 | 0.313 | 1 | 0.154 | 0.31 | 0.988 | 0.779 | 0.267 | 0.981 | 0.171 | 0.145 | 0.684 | 0.26 |
| R < 0 | 0.303 | 1 | 0.157 | 0.315 | 0.991 | 0.777 | 0.234 | 0.977 | 0.186 | 0.174 | 0.734 | 0.244 |
| R ≤ 0 | **0.054** | 0.433 | 0.626 | 0.351 | 0.166 | 0.767 | 0.145 | 0.901 | 0.344 | 0.432 | 0.256 | 0.471 |

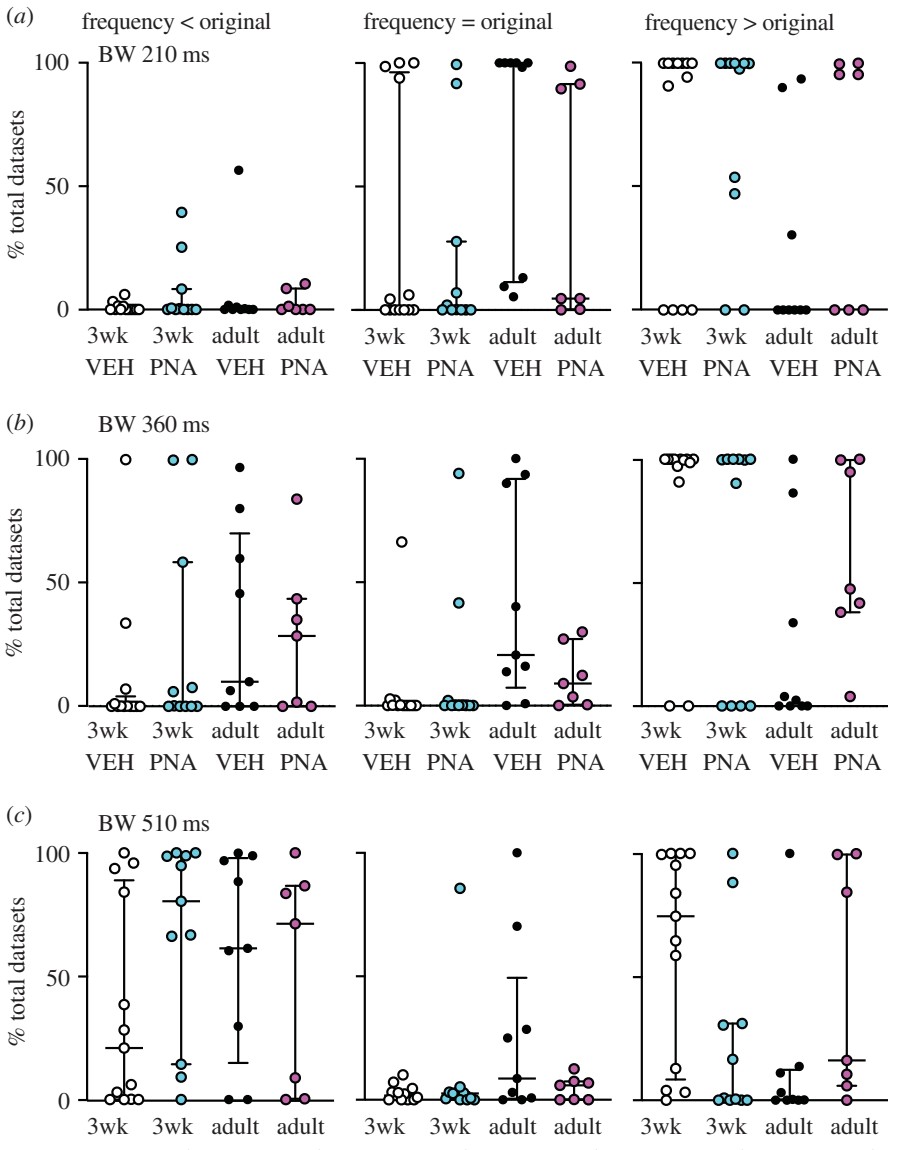

**Figure 6.** Burst frequency is lower in GnRH neurons in the physiologic range of burst windows (BW). Each dot shows the percentage of total datasets from an individual cell with a burst frequency less than (left), equal to (centre) or greater than (right) the original data's value for burst windows of (*a*) 210 ms, (*b*) 360 ms and (*c*) 510 ms. Graphs show individual values, median and interquartile range.

release. Cluster does not rely on hormone half-life to establish peaks, thus is appropriate for time-series data, such as firing rate, that do not have a half-life component. When the original data from individual cells were compared with that cell's total data distribution, the contribution of GnRH neuron physiology to long-term firing patterns was evident. Specifically, original data from both the 3wk and adult vehicle groups featured fewer peaks of longer duration than their respective MC datasets. Such a shift *in vivo* would be expected to facilitate an episodic pattern versus more continuous release. Treatment with a continuous GnRH regimen downregulates pituitary response, essentially shutting off the downstream reproductive system [64]. This phenomenon has been used to develop long-acting GnRH analogues for treatment of conditions such as precocious puberty [65] and illustrates the critical nature of the episodic GnRH release pattern. Interestingly, the long-term patterns of individual cells are not different from their MC distribution in PNA mice. This may help explain the high-frequency LH release in these mice and potentially the same phenomenon in women with PCOS.

Cluster has been used to examine patterns in firing rate of GnRH neurons [29,31,32] and arcuate kisspeptin neurons [37], which are putative upstream drivers of episodic GnRH release [66,67]. In those studies, Cluster revealed increased frequency of firing peaks in castrated mice versus

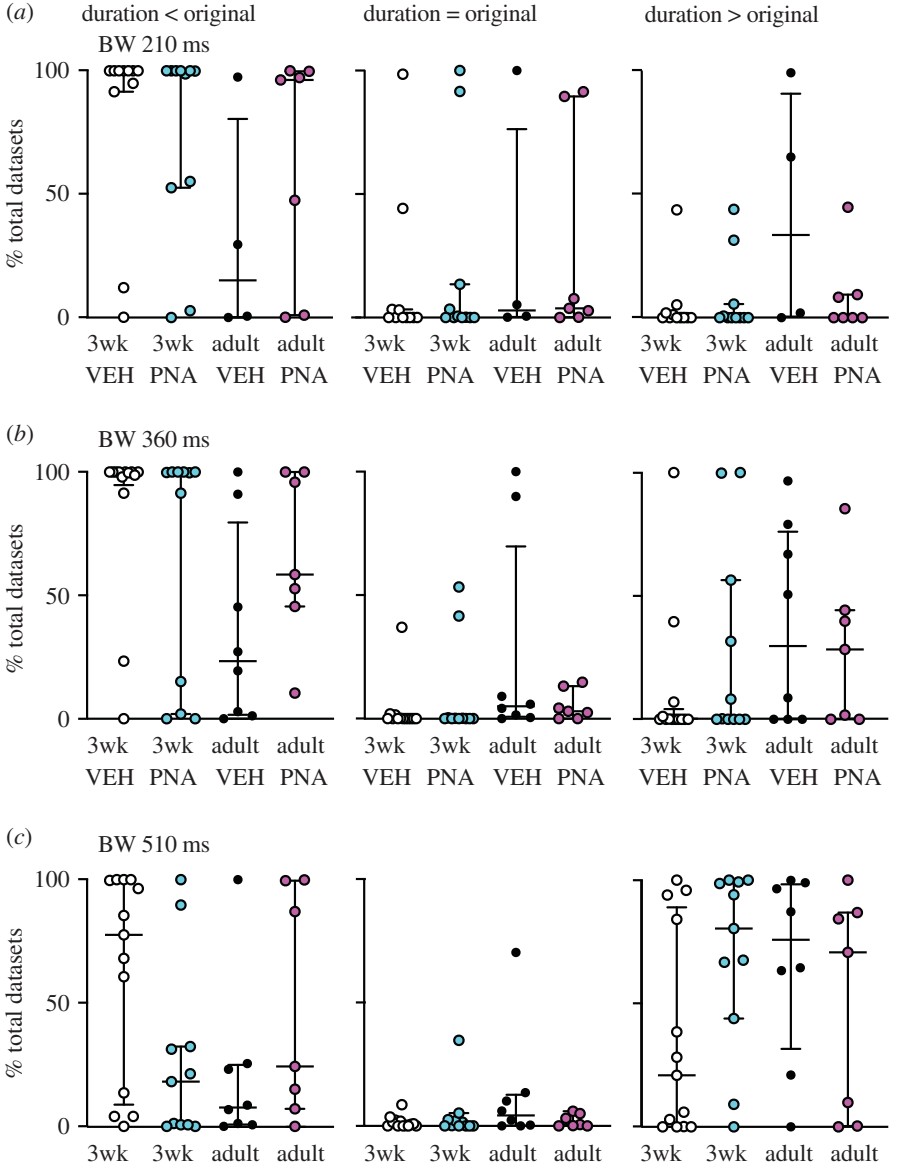

**Figure 7.** Burst duration is longer in GnRH neurons in the physiologic range of burst windows (BW). Each dot shows the percentage of total datasets from an individual cell with a burst duration that is less than (left), equal to (centre) or greater than (right) the original data's value for burst windows of (*a*) 210 ms, (*b*) 360 ms and (*c*) 510 ms. Graphs show individual values, median and interquartile range.

castrated mice in which homeostatic feedback was replaced via steroid implants, mirroring *in vivo* changes in LH pulses and validating this approach. Cluster analysis failed to reveal differences among groups in the present study when only the original data were examined. This may be attributable to the recordings being of shorter duration for the present study, precluding a rigorous estimation of long-term firing patterns from raw data alone and re-emphasizing the utility of the MC analysis.

The contribution of GnRH neuron physiology was also evident in analyses of short-term patterns, or bursting. Burst firing is thought to facilitate neurosecretion as repeated arrival of depolarizing action potentials in nerve terminals prolongs and enhances the rise in intracellular calcium required for vesicle fusion [38,68–71]. As with long-term patterns, GnRH neurons organize their bursts to be longer and less frequent than the MC dataset distributions. The exception was adult vehicle controls, for which burst frequency did not differ from the MC dataset distributions. This is also the only group examined that exhibits typical reproductive cycles, suggesting the hypothesis that maturation of GnRH neurons, and subsequent successful reproduction, involves a shift in short-term burst organization. The median percentage of MC datasets with parameter values less than or greater than the

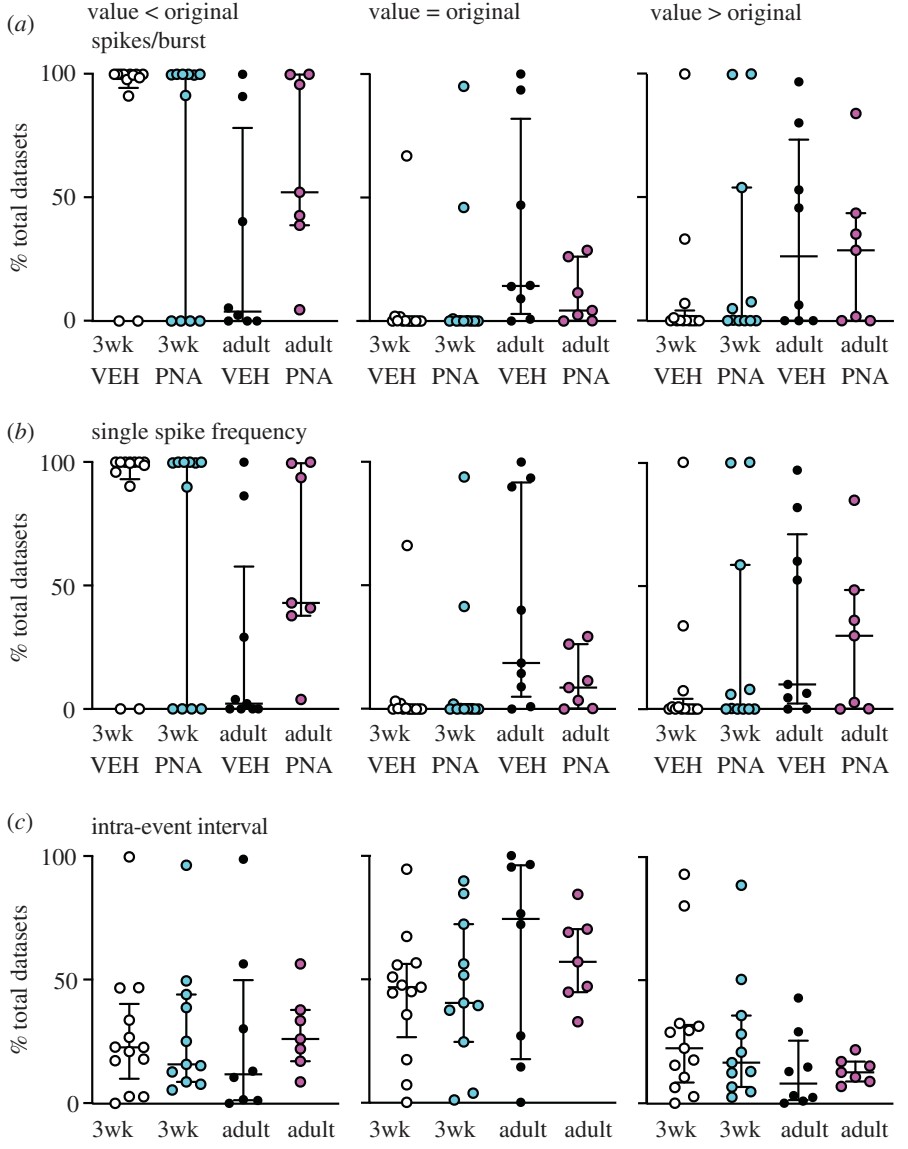

**Figure 8.** Comparison of original data to total datasets for selected burst parameters. Each dot shows the percentage of total datasets from an individual cell that is less than (left), equal to (centre) or greater than (right) the original data's value for the indicated parameter for a burst window of 360 ms. (a) spikes/burst; (b) single-spike frequency; (c) Intra-event interval. Graphs show individual values, median and interquartile range.

original data for adult controls is repeatedly different from that of three-week-old control mice. By contrast, values for adult PNA mice are intermediate between three-week-old and adult controls. Consistent with this, pairwise comparisons of groups at the physiologic burst windows revealed differences with maturation between three-week-old and adult controls, and differences with treatment in adults between control and PNA mice. The failure of adult PNA mice to undergo a similar transition suggests PNA treatment programmes a failure of maturation of the reproductive neuroendocrine system.

These observations are of interest with regard to LH release relative to sleep stage and how this relationship changes in women with PCOS and with puberty. Women with PCOS [72] and normal (male and female) pubertal subjects [73,74] exhibit similar relationships between sleep stages and LH pulse initiation. Specifically, LH pulses are more typically associated with slow-wave sleep patterns. By contrast, in normal mature mid–late follicular phase women, LH pulse initiation is more often preceded by increased wake episodes and fewer REM epochs compared with randomly selected time points. Wake and REM were not associated with LH pulse initiation in women with PCOS [72]. LH pulse initiation in the early follicular phase of the cycle is also preceded by brief awakenings [75].

**Table 5.** *p*-values generated by pairwise group permutation tests of median values for pairwise comparisons between groups on bursts detected by vary burst window. A, adult; 3wk, three week old. Italics $p \leq 0.05$; bold $0.05 < p < 0.1$.

| burst window → | 210 ms | | | | | 360 ms | | | | |
|---|---|---|---|---|---|---|---|---|---|---|
| parameter ↓, relation of randomized (R) to original (0) data → | R = 0 | R > 0 | R ≥ 0 | R < 0 | R ≤ 0 | R = 0 | R > 0 | R ≥ 0 | R < 0 | R ≤ 0 |
| **burst frequency** | | | | | | | | | | |
| A PNA versus A VEH | *0.002* | *0.017* | 1 | 1 | *0.01* | 0.426 | 0.142 | 1 | 1 | 0.153 |
| 3wk PNA versus A PNA | 0.481 | 0.528 | 1 | 1 | 0.485 | **0.088** | 0.539 | 0.278 | 0.263 | 0.509 |
| 3wk VEH versus A VEH | *0.007* | **0.078** | 1 | 1 | **0.068** | *0.019* | *0.033* | 0.066 | **0.058** | *0.046* |
| 3wk VEH versus 3wk PNA | 1 | 1 | 1 | 1 | 1 | 1 | 0.74 | 0.369 | 0.373 | 0.761 |
| **burst duration** | | | | | | | | | | |
| A PNA versus A VEH | 0.886 | **0.09** | 0.234 | 0.217 | **0.082** | 0.568 | 1 | 0.276 | 0.272 | 1 |
| 3wk PNA versus A PNA | 0.3 | 0.64 | 0.465 | 0.433 | 0.604 | 0.13 | 0.533 | 0.508 | 0.534 | 0.531 |
| 3wk VEH versus A VEH | 0.321 | *0.028* | **0.058** | *0.049* | *0.034* | *0.022* | *0.02* | *0.017* | *0.024* | *0.028* |
| 3wk VEH versus 3wk PNA | 0.662 | 1 | 0.698 | 0.669 | 1 | 1 | 0.355 | 0.765 | 0.736 | 0.352 |
| **spikes/burst** | | | | | | | | | | |
| A PNA versus A VEH | 0.626 | 0.283 | 0.412 | 0.414 | 0.269 | 0.364 | 0.995 | 0.202 | 0.22 | 0.991 |
| 3wk PNA versus A PNA | 0.473 | 0.641 | 0.472 | 0.42 | 0.644 | **0.089** | 0.285 | 0.52 | 0.543 | 0.296 |
| 3wk VEH versus A VEH | **0.091** | *0.027* | **0.062** | **0.085** | *0.032* | *0.024* | *0.035* | *0.016* | *0.025* | *0.036* |
| 3wk VEH versus 3wk PNA | 0.667 | 1 | 0.666 | 0.669 | 1 | 1 | 0.384 | 0.746 | 0.741 | 0.365 |
| **single-spike frequency** | | | | | | | | | | |
| A PNA versus A VEH | *0.013* | 0.379 | 0.109 | 0.107 | 0.397 | 0.45 | 1 | 0.156 | 0.149 | 1 |
| 3wk PNA versus A PNA | 0.432 | 0.235 | 0.483 | 0.53 | 0.221 | 0.08 | 0.295 | 0.38 | 0.364 | 0.31 |
| 3wk VEH versus A VEH | *0.005* | 1 | **0.072** | **0.066** | 1 | *0.016* | *0.047* | *0.029* | *0.036* | *0.045* |
| 3wk VEH versus 3wk PNA | 1 | 1 | 1 | 1 | 1 | 1 | 0.612 | 0.717 | 0.746 | 0.59 |

(*Continued.*)

**Table 5.** (*Continued.*)

| burst window → | 510 ms | | | | |
| Parameter ↓, relation of randomized (R) to original (O) data → | R = 0 | R > 0 | R ≥ 0 | R < 0 | R ≤ 0 |
| --- | --- | --- | --- | --- | --- |
| burst frequency | | | | | |
| A PNA versus A VEH | 0.684 | **0.057** | 1 | 1 | **0.082** |
| 3wk PNA versus A PNA | 0.132 | 0.534 | 1 | 1 | 0.477 |
| 3wk VEH versus A VEH | *0.026* | **0.069** | 0.329 | 0.368 | **0.06** |
| 3wk VEH versus 3wk PNA | 0.82 | *0.043* | 0.261 | 0.262 | **0.052** |
| burst duration | | | | | |
| A PNA versus A VEH | 0.386 | 0.867 | 0.304 | 0.263 | 0.842 |
| 3wk PNA versus A PNA | 1 | 1 | 0.815 | 0.809 | 1 |
| 3wk VEH versus A VEH | *0.032* | 0.205 | **0.098** | 0.1 | 0.21 |
| 3wk VEH versus 3wk PNA | 1 | 0.166 | 0.162 | 0.174 | 0.184 |
| spikes/burst | | | | | |
| A PNA versus A VEH | 0.645 | 0.834 | 0.202 | 0.198 | 0.861 |
| 3wk PNA versus A PNA | 1 | 1 | 0.411 | 0.373 | 1 |
| 3wk VEH versus A VEH | *0.03* | 0.243 | **0.086** | **0.082** | 0.218 |
| 3wk VEH versus 3wk PNA | 0.754 | 0.259 | **0.043** | **0.05** | 0.275 |
| single-spike frequency | | | | | |
| A PNA versus A VEH | 1 | 1 | **0.054** | **0.074** | 1 |
| 3wk PNA versus A PNA | 0.722 | 1 | 0.529 | 0.513 | 1 |
| 3wk VEH versus A VEH | **0.068** | 0.35 | **0.054** | **0.056** | 0.35 |
| 3wk VEH versus 3wk PNA | 0.822 | 0.243 | **0.053** | **0.054** | 0.241 |

LH pulse initiation in women with PCOS is thus more comparable to the immature state, as are GnRH neuron burst parameters in adult PNA mice in the present study.

Differences in burst parameters among groups were dependent upon the burst window chosen to define bursts. Given there are more bursts at the 360 ms burst window, intentionally set near the peak, increased observation of differences could reflect an increase in observations, thus statistical power. The MC approach should minimize this caveat by repeatedly permuting the intervals. Further arguing against the number of bursts contributing to the findings, low *p*-values were also observed across many categories for the 210 ms burst window, in which fewer bursts are detected. Of note, the increase in number of bursts detected when moving from the 210 to 360 ms burst window was similar to the decrease in the number of bursts detected when moving from the 360 to 510 ms burst window; despite a similar number of bursts, differences were typically not observed at 510 ms. This supports the concept that physiologic ranges for GnRH burst generation are more appropriate for these analyses.

The phenomena that generate episodic GnRH release, the 'GnRH pulse generator' are still not understood, although evidence is mounting for a possible location within the arcuate nucleus kisspeptin neurons [37,66,76]. The present work confirms action potential firing patterns of GnRH neurons in coronal brain slices, which would not be subject to ongoing input from this region, exhibit both long- and short-term patterns that change with age and a disease model. These observations suggest the biologic state of these cells contributes considerably to their output patterns. Understanding the intrinsic and synaptic properties that underlie this biology and how important inputs such as a pulse generator sculpt these properties are topics of interest for future studies.

Ethics. The work in this manuscript did not require approval by any ethics board as it was entirely *in silico*. We did include an approval statement for the collection of the original data.

Data accessibility. Analysis code: https://gitlab.com/um-mip/mc-project-code. Data used: https://deepblue.lib.umich.edu/data/concern/data_sets/xs55mc12w?locale=en.

Authors' contributions. J.P. carried out the Monte Carlo analysis and wrote associate code, performed data analysis, participated in the design of the study and drafted the manuscript; R.A.D. helped with code, participated in the design and edited the manuscript; E.A.D. collected the original data, participated in analysis and edited the manuscript; S.S. participated in the analysis and edited the manuscript; S.M.M. performed data analysis, participated in the design of the study and finalized the manuscript. All authors gave final approval for publication.

Competing interests. We declare we have no competing interests

Funding. This study was supported by NIH/Eunice Kennedy Shriver National Institute of Child Health and Human Development R37 HD34860 and P50 HD28934.

Acknowledgements. We thank Ms. Elizabeth Wagenmaker for editorial comments.

Disclosure statement. The authors have nothing to disclose.

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
