## [Reviewer comments · Royal Society Open Science]

Review History

RSOS-201040.R0 (Original submission)

Review form: Reviewer 1 (Roberto Maggi)

Is the manuscript scientifically sound in its present form?

Yes

Are the interpretations and conclusions justified by the results?

Yes

Is the language acceptable?

Yes

Do you have any ethical concerns with this paper?

No

Have you any concerns about statistical analyses in this paper?

No

Recommendation?

Accept as is

Comments to the Author(s)

This reviewer thanks the authors to have modified the manuscript according to the suggestions. There are not further comments.

Review form: Reviewer 2**Is the manuscript scientifically sound in its present form?**

Yes

Are the interpretations and conclusions justified by the results?

Yes

Is the language acceptable?

Yes

Do you have any ethical concerns with this paper?

No

Have you any concerns about statistical analyses in this paper?

No

Recommendation?

Accept with minor revision (please list in comments)

Comments to the Author(s)

The authors have mostly addressed the questions in my previous review. I have several suggestions that I believe would further improve the clarity of the Materials and Methods section.

1. On lines 101-102 the authors state, 'Using the surrogate data, we are then able to test the null hypothesis that the firing interval observed is generated by chance.' It is not clear if this hypothesis refers to intra- or inter- burst intervals or both. Could you please clarify. furthermore, as I have already mentioned above there are two types of events considered in this study, namely spikes and bursts, and respectively there are intra- and inter burst intervals. It is not clear if both types of intervals were subject to permutations? Please clarify and justify.
2. On lines 144-145 the authors state, 'Cluster and VBW algorithms were run as described on each of the 1000 MC datasets, yielding frequency distributions for the output parameters.' Could you please specify the output parameter for clarity.
3. Lines 169-170, the authors state, ' Permutation tests were thus used to determine if two sets of data have different distributions.' Could you please clarify distributions of what? Do you mean distributions of intervene intervals here?
4. On lines 194-196, the authors state, 'In this design, we test the null hypotheses that age and/or treatment does not affect the proportion of MC datasets that are $<$, $>$, $=$, \leq , \geq the corresponding original data.' Could you please clarify by specifying here what exactly is being compared between the MC datasets and the original data?

Decision letter (RSOS-201040.R0)

Dear Dr Moenter

On behalf of the Editors, I am pleased to inform you that your Manuscript RSOS-201040 entitled "Firing patterns of gonadotropin-releasing hormone (GnRH) neurones are sculpted by their biologic state" has been accepted for publication in Royal Society Open Science subject to minor revision in accordance with the referee suggestions. Please find the referees' comments at the end of this email.

The reviewers and handling editors have recommended publication, but also suggest some minor revisions to your manuscript. Therefore, I invite you to respond to the comments and revise your manuscript.

- Ethics statement

- Data accessibility

<http://datadryad.org/submit?journalID=RSOS&manu=RSOS-201040>

- Competing interests

- Authors' contributions

- Acknowledgements

- Funding statement

Because the schedule for publication is very tight, it is a condition of publication that you submit the revised version of your manuscript before 15-Jul-2020. Please note that the revision deadline will expire at 00.00am on this date. If you do not think you will be able to meet this date please let me know immediately.

- 1) A text file of the manuscript (tex, txt, rtf, docx or doc), references, tables (including captions) and figure captions. Do not upload a PDF as your "Main Document";
- 2) A separate electronic file of each figure (EPS or print-quality PDF preferred (either format should be produced directly from original creation package), or original software format);
- 3) Included a 100 word media summary of your paper when requested at submission. Please ensure you have entered correct contact details (email, institution and telephone) in your user account;
- 4) Included the raw data to support the claims made in your paper. You can either include your data as electronic supplementary material or upload to a repository and include the relevant doi within your manuscript. Make sure it is clear in your data accessibility statement how the data can be accessed;
- 5) All supplementary materials accompanying an accepted article will be treated as in their final form. Note that the Royal Society will neither edit nor typeset supplementary material and it will

be hosted as provided. Please ensure that the supplementary material includes the paper details where possible (authors, article title, journal name).

If your manuscript is newly submitted and subsequently accepted for publication, you will be asked to pay the article processing charge, unless you request a waiver and this is approved by Royal Society Publishing. You can find out more about the charges at <https://royalsocietypublishing.org/rsos/charges>. Should you have any queries, please contact opencience@royalsociety.org.

Kind regards,
Andrew Dunn
Royal Society Open Science Editorial Office
Royal Society Open Science
opencience@royalsociety.org

on behalf of Prof Kevin Padian (Subject Editor)
opencience@royalsociety.org

Editor comments:

Thanks for your attention to requested revisions. We are pleased to accept your article with a few modifications. We appreciate your submission to RSOS.

Associate Editor Comments to Author:

The referees have a few minor changes to implement, but once these have been incorporated, the paper may be accepted.

Reviewer comments to Author:

Reviewer: 1

Comments to the Author(s)

This reviewer thanks the authors to have modified the manuscript according to the suggestions. There are not further comments.

Reviewer: 2

Comments to the Author(s)

The authors have mostly addressed the questions in my previous review. I have several suggestions that I believe would further improve the clarity of the Materials and Methods section.

1. On lines 101-102 the authors state, 'Using the surrogate data, we are then able to test the null hypothesis that the firing interval observed is generated by chance.' It is not clear if this hypothesis refers to intra- or inter- burst intervals or both. Could you please clarify. furthermore, as I have already mentioned above there are two types of events considered in this study, namely spikes and bursts, and respectively there are intra- and inter burst intervals. It is not clear if both types of intervals were subject to permutations? Please clarify and justify.
2. On lines 144-145 the authors state, 'Cluster and VBW algorithms were run as described on each of the 1000 MC datasets, yielding frequency distributions for the output parameters.' Could you please specify the output parameter for clarity.
3. Lines 169-170, the authors state, ' Permutation tests were thus used to determine if two sets of data have different distributions.' Could you please clarify distributions of what? Do you mean distributions of intervene intervals here?
4. On lines 194-196, the authors state, 'In this design, we test the null hypotheses that age and/or treatment does not affect the proportion of MC datasets that are $<$, $>$, $=$, \leq , \geq the corresponding original data.' Could you please clarify by specifying here what exactly is being compared between the MC datasets and the original data?

Author's Response to Decision Letter for (RSOS-201040.R0)

See Appendix A.

Decision letter (RSOS-201040.R1)

Dear Dr Moenter,

It is a pleasure to accept your manuscript entitled "Firing patterns of gonadotropin-releasing hormone (GnRH) neurones are sculpted by their biologic state" in its current form for publication in Royal Society Open Science. The comments of the reviewer(s) who reviewed your manuscript are included at the foot of this letter.

You can expect to receive a proof of your article in the near future. Please contact the editorial office (openscience_proofs@royalsociety.org) and the production office (openscience@royalsociety.org) to let us know if you are likely to be away from e-mail contact -- if

you are going to be away, please nominate a co-author (if available) to manage the proofing process, and ensure they are copied into your email to the journal.

on behalf of Prof Kevin Padian (Subject Editor)
openscience@royalsociety.org

Appendix A

We thank the editors and reviewers for their comments and have provided a point-by-point response below in blue

Reviewer: 2

Comments to the Author(s)

The authors have mostly addressed the questions in my previous review. I have several suggestions that I believe would further improve the clarity of the Materials and Methods section.

1. On lines 101-102 the authors state, 'Using the surrogate data, we are then able to test the null hypothesis that the firing interval observed is generated by chance.' It is not clear if this hypothesis refers to intra- or inter- burst intervals or both. Could you please clarify. furthermore, as I have already mentioned above there are two types of events considered in this study, namely spikes and bursts, and respectively there are intra- and inter burst intervals. It is not clear if both types of intervals were subject to permutations? Please clarify and justify.

Thank you for your comment. We believe the confusion stems from our referring to intervals, which is what was shuffled in the present work, and the reviewer (and likely other readers) referring to events. We have edited this text to read

“Using the surrogate data, we are then able to test the null hypothesis that the firing intervals and the resulting groupings of events observed (individual spikes, bursts, clusters) are generated by chance.”

2. On lines 144-145 the authors state, 'Cluster and VBW algorithms were run as described on each of the 1000 MC datasets, yielding frequency distributions for the output parameters.' Could you please specify the output parameter for clarity.

Parameters have been listed in a parenthetical inclusion as requested. (Cluster: frequency amplitude and duration of elevated firing; VBW: burst frequency, interevent and intraevent intervals, burst duration, spikes/burst, single spike frequency).

3. Lines 169-170, the authors state, ' Permutation tests were thus used to determine if two sets of data have different distributions.' Could you please clarify distributions of what? Do you mean distributions of intervene intervals here?

We examined distributions of all the parameters identified by both Cluster and VBW analysis as identified in comment 2. We have added this to the text.

4. On lines 194-196, the authors state, 'In this design, we test the null hypotheses that age and/or treatment does not affect the proportion of MC datasets that are $<$, $>$, $=$, \leq , \geq the corresponding original data.' Could you please clarify by specifying here what exactly is being compared between the MC datasets and the original data?

We examined original data vs distributions of all the parameters identified by both Cluster and VBW analysis as identified in comment 2. We have added this to the text.